# 3D Bioprinting of an Endothelialized Liver Lobule-like Construct as a Tumor-Scale Drug Screening Platform

**DOI:** 10.3390/mi14040878

**Published:** 2023-04-19

**Authors:** Zicheng Fan, Xiaoyun Wei, Keke Chen, Ling Wang, Mingen Xu

**Affiliations:** 1School of Automation, Hangzhou Dianzi University, Hangzhou 310018, China; fzc@hdu.edu.cn (Z.F.); kkchen@hdu.edu.cn (K.C.); 2Key Laboratory of Medical Information and 3D Bioprinting of Zhejiang Province, Hangzhou Dianzi University, Hangzhou 310018, China

**Keywords:** 3D bioprinting, GelMA hydrogel, liver lobule-like construct, drug screening

## Abstract

3D cell culture models replicating the complexity of cell–cell interactions and biomimetic extracellular matrix (ECM) are novel approaches for studying liver cancer, including in vitro drug screening or disease mechanism investigation. Although there have been advancements in the production of 3D liver cancer models to serve as drug screening platforms, recreating the structural architecture and tumor-scale microenvironment of native liver tumors remains a challenge. Here, using the dot extrusion printing (DEP) technology reported in our previous work, we fabricated an endothelialized liver lobule-like construct by printing hepatocyte-laden methacryloyl gelatin (GelMA) hydrogel microbeads and HUVEC-laden gelatin microbeads. DEP technology enables hydrogel microbeads to be produced with precise positioning and adjustable scale, facilitating the construction of liver lobule-like structures. The vascular network was achieved by sacrificing the gelatin microbeads at 37 °C to allow HUVEC proliferation on the surface of the hepatocyte layer. Finally, we used the endothelialized liver lobule-like constructs for anti-cancer drug (Sorafenib) screening, and stronger drug resistance results were obtained when compared to either mono-cultured constructs or hepatocyte spheroids alone. The 3D liver cancer models presented here successfully recreate liver lobule-like morphology, and may have the potential to serve as a liver tumor-scale drug screening platform.

## 1. Introduction

Cancer poses a serious threat to human health due to its increasing incidence and high mortality [1,2]. Liver cancer, as one of the most common malignant tumors, has become the third leading cause of cancer death worldwide [3,4]. Currently, only one drug, Sorafenib, is approved for liver cancer treatment [5]. Anti-tumor drug screening is an important part in the discovery and development of new drugs for liver cancer treatment. Conventional 2D culturing is widely used in drug screening due to its low cost and ease of reproduction. However, there are significant differences between 2D cultures and in vivo environments that can impact the expression of key genes during drug reactions [6,7,8]. Animal models have long been regarded as the gold standard for drug testing; however, the physiological differences between species are not negligible [9,10]. Hence, 3D hepatocyte culture has been regarded as a promising method for developing complex liver cancer models that can recapitulate the tumor microenvironment and drug response [11,12,13].

Scaffold-free liver cancer spheroids are among the simplest forms of 3D culture models. Cells in suspension are self-organized into spheroids by means of hanging drop, microwell array, and magnetic assembly [14]. Cancer spheroids provide tight cell–cell and cell–extracellular matrix (ECM) interactions that mimic solid tumors, exhibiting strong capacity for drug resistance similar to in vivo tumor tissue [15,16]. Nevertheless, spheroids are limited to small-scale models, normally around several hundred micrometers, which is far from the scale of native tumors (on the scale of millimeters to centimeters) [17]. By combining cells and hydrogel through 3D bioprinting technology, 3D liver cancer models with more complex structures and tumor-scale microenvironments can be constructed [18,19,20]. Many hepatic tumor microtissues have been reported by depositing cell-laden hydrogels; however, such models cannot thoroughly recapitulate human liver architecture, and as such have poor reliability for liver disease or hepatotoxicity studies.

In native tumor tissues, interaction between tumor cells and endothelial cells (ECs) directly affects the transport of nutrients and metabolites [21,22,23]. For instance, ECs co-cultured with HepG2 cells in a 3D culture system are able to successfully establish vascular networks, and have shown stronger cellular functions and activities in contrast to 2D cultures [24]. In another example, a 3D endothelialized hepatic tumor microtissue model was fabricated by co-culturing ECs with human HepG2 cells within porous microspheres, which exhibited significantly higher half-maximal inhibitory concentration values against anti-cancer drugs [25]. Thus, in vitro liver cancer models with ECs for the establishment of vascular networks are vital for the evaluation of drug targets.

In the body, the liver is a highly vascularized and heterogeneous organ; liver lobules are the basic unit, and each lobule is composed of parenchymal cells (hepatocytes), non-parenchymal cells (such as epithelial cells and endothelial cells), and the intrahepatic vascular system [26,27]. To replicate this complex heterogeneous environment and provide biologically-relevant size, we fabricated a liver tumor-scale model with similar structural morphology to the liver lobules using our previously developed dot extrusion printing (DEP) technology. DEP technology possesses the capacity to produce uniform hydrogel microbeads with precise localization [28]. In this work, a millimeter-sized hexagonal structure was developed by printing gelatin methacrylate (GelMA) hydrogel microbeads encapsulating C3A cells, a type of human hepatocellular carcinoma cell lines, to mimic the functionality of lobule-like units. Further, we printed human umbilical vein endothelial cell (HUVEC) components onto the C3A layer, improving the bio-imitability of the liver cancer model. On co-culturing the C3A cells and HUVECs within the 3D lobule-like structures, adjacent HUVECs were interconnected, ultimately elongating and branching to form a layer of vascular network structure on the surface of the constructs. Finally, the engineered tumor-scale liver lobule-like constructs were employed to assess the cell responses of the chemotherapeutic anti-cancer drug Sorafenib and the effects on tumor growth and viability were analyzed for the co-culture model, mono-culture model, liver cancer spheroids.

## 2. Materials and Methods

### 2.1. Bioprinting Platform

In this work, the 3D bioprinter (Bio Architect WS) we used during the whole experiment was obtained from Hangzhou Regenovo Biotechnology Co., Ltd (Hangzhou, China). This bioprinter mainly contains four standardized channel interfaces with four kinds of printheads loaded, an X–Y–Z moving part, and a dual temperature control unit to regulate the temperature of the printhead and the receiving platform. The movement of the X–Y axis is controlled by the sliding table, while the Z-axis is controlled by the rise and fall of the platform.

### 2.2. Bioprinting of Lobule-Like Structures

In order to construct a model with a bionic structure, a pre-designed G code was used to print hydrogel microbeads along a print path of a hexagonal structure. Different concentrations of GelMA prepolymer solution, including 6%, 8%, and 10% (*w*/*v*) with 0.5% (*w*/*v*) lithium phenyl-2, 4, 6-trimethylbenzoylphosphinate (LAP, Sigma Aldrich) were prepared in advance. Briefly, phosphate buffered saline (PBS, Gibco, Thermo Fisher Scientifific, Waltham, MA, USA) was used as a solvent to dissolve the corresponding volume of GelMA material with LAP. Then, GelMA solution was transferred to a 5 mL syringe and loaded into the 3D bioprinter. The pneumatic pressure and dispensing time were respectively fixed at 100 kPa and 1000 ms to ensure that the size of the printed microbeads was around 700 μm according to our previous results [28]. The center-to-center distance between microbeads was set as 700 μm to ensure connection between two adjacent microbeads. As For printing of GelMA with different concentrations, the temperature of the printhead was set between 10 and 20 °C to ensure good formability and printability of the bioink.

### 2.3. SEM Characterization of GelMA Hydrogel and Porosity Analysis

Three different concentrations of bulk hydrogel (2 × 2 × 0.5 cm) were prepared. Specifically, after photocrosslinking, the hydrogel bulk samples were frozen in a refrigerator at −80 °C for 2 h and then placed in a lyophilizer for freeze-drying. Samples were sputtered with gold and then observed and photographed with a scanning electron microscope (SEM, JSM-6460, JEOL, Tokyo, Japan). The average pore sizes of the samples were measured using ImageJ software.

### 2.4. Mechanical Testing

Hydrogel structures with a height of 10 mm were prepared in advance. The compression moduli of hydrogels scaffolds in different concentrations were measured using a mechanical testing instrument (Instron, 5943, Boston, MA, USA). All hydrogel structures were tested at the same compression rate (1 mm/min) and compressed to 50% strain level. The compression modulus was calculated from the slope of the linear interval of the stress–strain curve (0–20% strain).

### 2.5. Cell Culturing

The C3A cells and HUVECs used in this work were purchased from Beijing Beina Chuanglian Biotechnology Institute, China. C3A cells were cultured in high-glucose Dulbecco’ s Modified Eagle’s Medium (DMEM, Gibco) supplemented with 10% (*v*/*v*) fetal bovine serum (FBS, Gibco) and 1% (*v*/*v*) penicillin/streptomycin (Gibco), while HUVECs were cultured using endothelial cell medium. Cells were maintained in an incubator at 37 °C with 5% CO_2_. Cells with 80~90% confluency were sub-cultured after dissociation with 0.125% trypsin. The medium was changed every other day.

### 2.6. Fabrication of Liver Lobule-Like Constructs

C3A cells with a density of 4 × 10^6^ mL^−1^ were respectively encapsulated in 6%/8%/10% (*w*/*v*) GelMA solution to prepare the bioink for printing. Fluorinated ethylene propylene (FEP) films were served as substrate to receive deposited hydrogel beads. After printing, the liver lobule-like constructs were exposed to 200 mW cm^−2^ UV light (405 nm) for 30 s and transferred to 12-well culture plates for a long-term culture; the culture medium was changed every other day.

### 2.7. Fabrication of Endothelialized Liver Lobule-Like Constructs

HUVECs with a density of around 2 × 10^6^ mL^−1^ were pre-encapsulated in 4% (*w*/*v*) gelatin solution and loaded into another printhead on standby. After the first C3A layer was fabricated, we switched the printhead to print the gelatin containing HUVECs directly onto the C3A layer by raising the Z-coordinate of the printhead by 300 μm. Afterwards, the co-cultured constructs were exposed to 200 mW cm^−2^ UV light (405 nm) for 30 s for photocrosslinking. We then transferred the constructs to a 24-well plate and put it in the incubator for 20 min. After the sacrificial gelatin phase flowed away, medium was added for subsequent culture.

### 2.8. Evaluation of Cell Viability and Proliferation

A live/dead cell viability kit (Thermo Fisher Scientifific, Waltham, MA, USA) was used to access cell viability according to the manufacturer’s instructions at the specified time points. Live and dead cells were stained with calcein-AM and propidium iodide (PI), respectively. After 20 min of incubation at 37 °C, the live and dead cells were observed using an inverted fluorescence microscope (Nikon, Ti-U, Tokyo, Japan).

The number of live and dead cells was counted by ImageJ software using at least three images from different areas of three bioprinted constructs for each condition. Significantly, cell viabilities before and after printing were calculated by the ratio of the number of live cells to the total number of cells. After long-term culture, red and green fluorescent images of the samples were obtained and cell viability was determined by the ratio of the green area to the total area.

To characterize the proliferation and morphology of the C3A cells in the constructs over a two-week culture period, bright-field images of the constructs were captured on days 1, 7, and 14. We then recorded the diameter distribution and average diameter of cell clusters in the constructs at different time points using ImageJ software.

### 2.9. Cell Morphology Analysis

F-actin staining was performed on mono-cultured liver lobule-like constructs of 6% GelMA hydrogel on days 1, 7, and 14 of culture, respectively, together with the endothelialized liver lobule-like constructs on days 1 and 14 of culture. Briefly, the selected samples were first washed using PBS, then 4% paraformaldehyde (Beyotime, Shanghai, China) was used to fix the samples for 4 h. After fixation, samples were washed and stained with Alexa Fluor 488 phalloidin (Invitrogen, Carlsbad, CA, USA) applied at 1:200 dilution for 2 h at room temperature. Finally, DAPI (Biosharp, Beijing, China) solution at 1:1000 in PBS was performed for nuclear detection for 10 min. After staining, all samples were washed thrice with PBS and observed using a confocal microscope (Nikon, A1RHD25, Tokyo, Japan).

### 2.10. Drug Treatment

Both co-cultured and mono-cultured liver lobule-like constructs were cultured for 14 days before drug treatment. Sorafenib powder (Sigma, St. Louis, MO, USA) was dissolved in 10 mM mL^−1^ stock solution, then 2.5 μL, 5 μL, 7.5 μL, and 10 μL of stock solution was respectively added to 1 mL medium to prepare 25, 50, 75, and 100 μM mL^−1^ working fluids. The co-cultured, mono-cultured, and prepared C3A spheroids were treated with the working fluids. Non-drug treatment samples were observed as a control. After 48 h of drug incubation, the samples were washed with PBS and then used for cell viability analysis.

### 2.11. Statistical Analysis

Each group of experiments was repeated at least three times. The experimental data are usually expressed as mean ± standard deviation. GraphPad Prism and imageJ software were used to statistically analyze the data, and Inkscape was used to draw the charts. The data were analyzed with *t* tests, with *p* < 0.05 indicating a significant difference.

## 3. Results and Discussion

### 3.1. Patterning Hydrogel Microbeads to Engineer Lobule-Like Structures

In this work, GelMA hydrogel microbeads were printed using DEP technology to build lobule-like structures (Figure 1A); the main printing processes are displayed in Figure 1B. According to our previous study [28], we produced GelMA hydrogel microbeads with a scale of 700 μm in diameter using optimized printing parameters, including driving pressure and dispensing time. Connection of the microbeads was achieved by regulating their center distance, thereby achieving lobule-like structures in an effective manner. Further, to explore the ability of our DEP system to accurately deposit and print multi-layer structures, we tried to print different layers of lobule-like structures; the structural morphology is shown in Figure 1C. The height of printed constructs with different layers increased linearly (Figure 1D), indicating that the printed multi-layer structures could maintain good structure without microbead collapse. With regard to the complex structure of multiple cell types in natural liver cancer tissue, we further verified the use of DEP technology to print heterogeneous lobule-like structures. The printing process was divided into the following parts: (i) printing hydrogel microbeads with one type of cells to fabricate the first lobule layer; (ii) switching to another printhead loaded with bioink encapsulating another type of cells and raising the Z-coordinate; and (iii) printing the second cell layer. Heterogeneous lobule-like structures with two layers were fabricated by printing of GelMA hydrogel stained with green dye and gelatin hydrogel stained with red dye (Appendix A). Therefore, the proposed DEP system exhibited its ability to print heterogeneous structures.

To determine the feasibility of our DEP technology for viable tissue construction, GelMA hydrogels of 6%/8%/10% (*w*/*v*) encapsulating C3A cells were printed. Then, we analyzed cell viabilities before and after printing. Most areas of the constructs showed green after immunofluorescence staining (Figure 1E). Cell viability after printing was slightly lower than before printing, and was around 90% in all three concentrations of hydrogel constructs (Figure 1F), demonstrating the high biocompatibility of DEP technology.

### 3.2. GelMA Hydrogel Characterization

ECM is an important part of the liver microenvironment, providing structural support and adhesion for resident cells [29,30,31]. Natural hydrogels such as alginate, chitosan, gelatin, and silk have been commonly used as ECM in liver tissue engineering, and have good biocompatibility and bioactivity [32,33]. However, natural hydrogels have fixed physical and chemical properties and poor mechanical strength, meaning that they lack flexibility in applications where different needs must be met. On the contrary, the physical and biochemical characteristics of synthetic hydrogels largely depend on their composition, polymerization method, and cross-linking density, providing greater coordination and mechanical strength in the manufacturing process [34]. GelMA is modified from gelatin; there are many binding sites distributed across all chains of the GelMA hydrogels, which promotes cell adhesion and allows cells to proliferate and migrate within GelMA hydrogel scaffolds [35,36,37].

GelMA hydrogel can be treated with freeze-drying to generate a porous structure [38,39,40,41]. In this work, the pores of GelMA hydrogels of different concentrations were measured by SEM. Cross-sections of hydrogels of all concentrations showed porous honeycomb structures, as shown in Figure 2A. The pores of GelMA hydrogels with different concentrations of 6%, 8%, and 10% (*w*/*v*) were quantitatively analyzed. The average pore diameters of the three hydrogels were 72.36 ± 12.7 μm, 108.87 ± 12.98 μm, and 138.22 ± 17.83 μm, respectively, indicating that pore size decreased as hydrogel concentration increased (Figure 2B).

The mechanical properties of GelMA hydrogel are the key factor in maintaining its long-term structural stability, and influence cell growth as well. To determine the influence of different concentrations of hydrogel on the mechanical properties of materials, we performed compression tests using 6%, 8%, and 10% (*w*/*v*) GelMA hydrogels. Consistent with the results reported previously [42], we found that there was a positive correlation between the hydrogel concentration and compression modulus (Figure 2C). Accordingly, 10% GelMA hydrogel showed the highest compression modulus around 23.05 ± 0.82 kPa, and the compression modulus of 8% and 6% GelMA hydrogel decreased successively to 14.23 ± 4.28 kPa and 10.95 ± 4.19 kPa, respectively (Figure 2D). These results enabled us to prepare hydrogel constructs with different mechanical properties by adjusting their concentrations.

### 3.3. Liver Lobule-Like Construct Fabrication and Culture

Furthermore, C3A cells were encapsulated in GelMA hydrogel to print 3D liver lobule-like constructs. To determine whether the three concentrations of hydrogels had a positive effect on the proliferation of cells, cell proliferation and viability were characterized during two weeks of culture, and bright-field, F-actin fluorescent, and live/dead cell staining images were captured at days 1, 7, and 14. As Figure 3A shows, after marking by F-actin dye, cells in the lobule-like constructs at day 1 were separately dispersed and emitted a faint fluorescence. Over the next 7 and 14 days, these cells proliferated to form spheroids. Next, we observed the viability and proliferation of cells in the three concentrations of hydrogel constructs. During two-week culture, cells maintained good viability (˃90%) at all three hydrogel concentrations, and the cells proliferated and formed large cell spheroids in hydrogel (Figure 3B,C). The diameter distribution of cell spheroids after 14 days of culturing was analyzed, with the results plotted in Figure 3D. We found that there was a much higher percentage of spheroids (diameter ˃ 60 μm) in the 6% GelMA construct as compared to the 8% and 10% GelMA constructs, indicating that 6% GelMA was most suitable for C3A culture.

It is worth noting that the cells encapsulated in the hydrogel maintained high viability for two weeks, and few dead cells were detected. The GelMA hydrogel we used here demonstrated good biocompatibility and provided a good mimetic ECM environment for cells under the premise of ensuring printability and structural stability. The 6% GelMA showed the best performance in terms of C3A cell spheroid growth, possibly due to the compression modulus of 6% GelMA (~10 kPa) being closer to that of liver tissue, as previously reported [43]. In addition, the most important factor is that the constructs adopt a bionic structure and have a hexagonal porous design rather than a closed hydrogel bulk, which makes it more convenient for all parts of the constructs to contact the culture medium, absorb nutrients, and discharge metabolic waste in time to avoid necrosis cores in the center of the model.

### 3.4. Development of Endothelialized Liver Lobule-Like Constructs

To mimic the complex liver tumor microenvironment, we co-cultured HUVECs and C3A cells for construction of an endothelialized liver cancer model. As shown in Figure 4A, endothelialized liver lobule-like constructs were developed in four steps by using two printheads. First, one type of bioink with 6% GelMA containing C3A cells was printed to fabricate the first layer. Then, another printhead loaded with bioink with 4% gelatin containing HUVECs was utilized to print cell-laden microbeads onto the first C3A layer. Afterwards, the two-layer lobule construct was photocured for GelMA crosslinking. Finally, a layer of HUVECs coating the liver lobule construct was achieved by sacrificing gelatin microbeads in the incubator for 20 min. We further performed fluorescence staining on the endothelialized lobule-like constructs to detect the network. As Figure 4B displays, after 14 days of co-culture the endothelial cells were fully extended and connected to form a thin network covering the surface of the C3A cells, as expected.

We successfully fabricated endothelialized liver lobule-like constructs using a combination of GelMA and gelatin microbead printing. The advantage of using these two materials is that GelMA is modified from gelatin, meaning that the two materials can be well integrated together to maintain structural stability. In addition, the GelMA surface has abundant cell attachment sites, which on the one hand ensures that HUVECs can quickly adhere to the model surface during the sacrificial phase, facilitating HUVEC attachment; on the other hand, the 3D structure guides HUVECs to extend along the surface of constructs, accelerating cell organization and network formation.

### 3.5. Drug Screening

Finally, we proceeded to explore the application of the endothelialized liver lobule-like constructs in drug screening. After 14 days, both co-cultured and mono-cultured models were treated with different concentrations of Sorafenib (25, 50, 75 and 100 µM) for 48 h. Cell viability was measured using a live/dead cell viability kit. From our results, under the low concentration of Sorafenib (25 μM) little difference was found between mono-cultured models and co-cultured models. At drug concentrations higher than 50 μM, the endothelialized liver lobule-like constructs showed higher cell viability, and cell viability was over 50% even under a high drug concentration of 100 μM, while the mono-liver lobule group showed a lesser resistance to Sorafenib than the co-cultured models (Figure 5A,B). In addition, C3A spheroids were prepared using an ultra-low attachment 96-well culture plate with U-bottom and subjected to further Sorafenib drug screening. Notably, after 48 h drug treatment, C3A spheroids began to respond to the drug at a low concentration of 10 μM, and were almost dead at drug concentrations of 50 μM (Appendix A).

Here, the developed 3D endothelialized liver lobule-like constructs replicating the key architecture of native liver tumors demonstrated stronger drug resistance compared to mono-liver models or 3D spheroids. We hypothesize that this might be due to the endothelialized liver lobule-like constructs having an endothelial barrier able to block the diffusion of Sorafenib to cells. In addition, we observed that Sorafenib triggered disruption of the endothelial network, which might be due to the ability of Sorafenib to inhibit angiogenesis, as reported in previous studies [23]. As a result, the printed endothelialized liver lobule-like constructs may serve as a tumor-scale model for drug efficacy screening.

## 4. Conclusions

In this study, a 3D liver cancer model with an endothelialized liver lobule-like architecture was accurately built utilizing our previously developed DEP technology, aiming to provide an in vitro liver cancer model with specific architecture at the tumor scale. We expanded DEP technology to print heterogeneous liver cancer models by printing C3A cells and HUVECs in GelMA and gelatin hydrogel, respectively, offering an efficient means of 3D co-culture model development. From our results, GelMA hydrogel at an appropriate concentration can provide a biomimetic ECM environment and promote the growth of hepatocytes while providing a suitable surface for endothelial cell adhesion and proliferation. Moreover, the developed models showed significantly enhanced efficacy of Sorafenib in comparison to that of mono-cultured liver cancer constructs or 3D C3A spheroids, which is probably because of the blocked diffusion of Sorafenib by the intact endothelial barrier structure. There remain several limitations to our current research. The native liver lobule unit (1 mm) is arranged in a hexagonal spatial architecture, with hepatocytes radially lining up to form hepatic cords and being separated by microvascular channels [44]. In this work, the bioprinted liver lobule-like constructs could only provide a similar morphology to the liver lobule, and lacked the vascular channels.

All in all, this DEP technology has potential applications in other 3D tumor models, where the combination of cancer cells and other stromal cells could provide better reproduction of complex tumor microenvironments. The fabricated endothelialized liver cancer models with large-scale liver lobule-like structures for drug testing may produce pharmacodynamic results that are closer to the actual conditions, providing an effective platform for in vitro anti-liver cancer drug testing.

## Figures and Tables

**Figure 1 micromachines-14-00878-f001:**
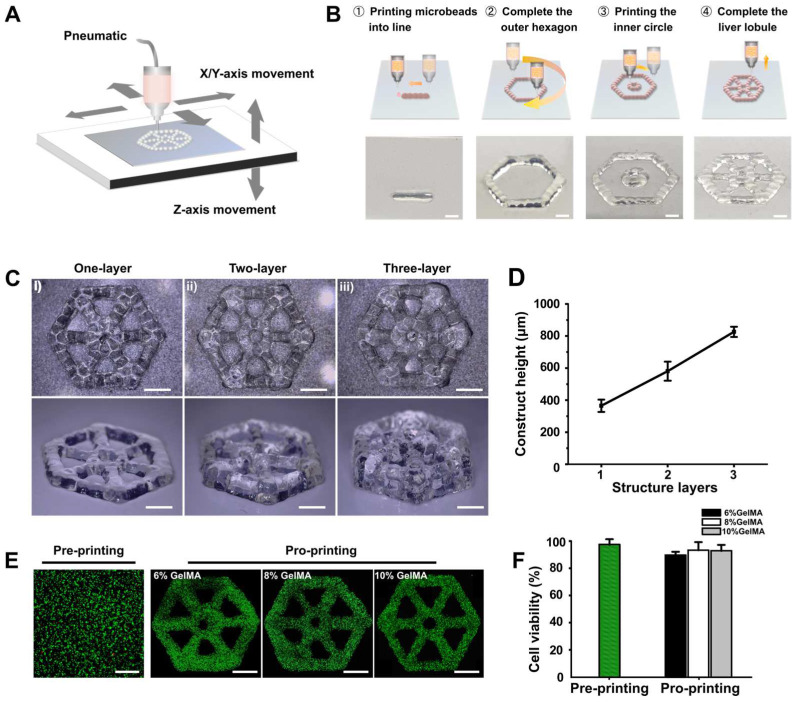
Fabrication of liver lobule-like constructs. (**A**) Schematic diagram showing the production of liver lobule-like structures using GelMA hydrogel beads generated by the DEP system. (**B**) Fabrication steps for liver lobule-like structures and corresponding images. Scale bar: 1 mm. (**C**) Images showing lobule-like structures with different layers. Scale bar: 1 mm. (**D**) Construct height of lobule-like structures with different layers. (**E**) Images showing live/dead analysis of C3A cells before and after printing. Scale bar: 1 mm. (**F**) Corresponding cell viabilities before and after printing.

**Figure 2 micromachines-14-00878-f002:**
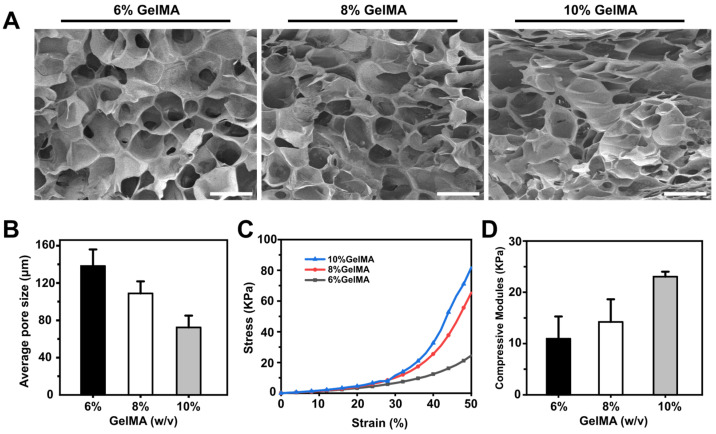
(**A**) Morphology characterization of GelMA hydrogels with concentrations of 6%, 8%, and 10% by SEM. Scale bar: 200 µm. (**B**) Quantitative analysis of the average pore size in the GelMA hydrogel of 6%, 8%, 10% concentrations. (**C**) Compressive stress–strain curves of GelMA hydrogels at different concentrations. (**D**) Compression modulus of hydrogels at three different concentrations of 6%, 8%, 10%.

**Figure 3 micromachines-14-00878-f003:**
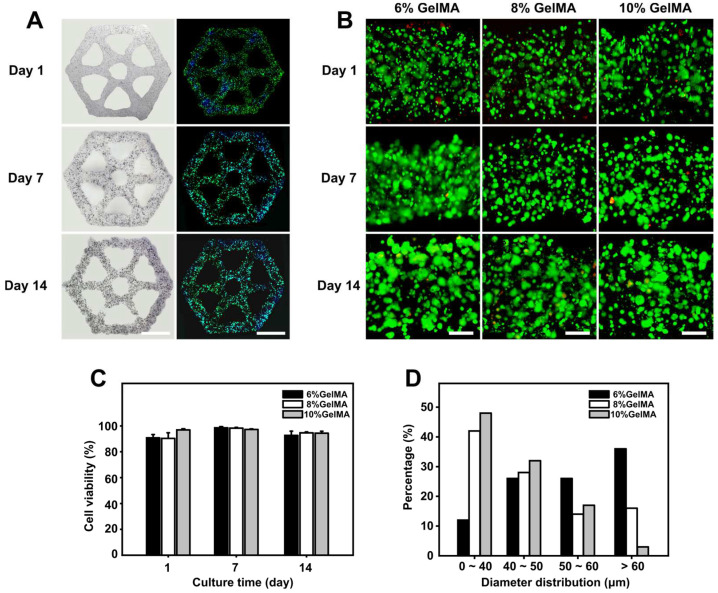
Liver lobule-like constructs fabrication and culture. (**A**) Bright-field and F-actin fluorescent images showing liver lobule-like constructs at days 1, 7, and 14. Scale bar: 2 mm. (**B**) Live/dead analysis of liver lobule-like constructs printed with different GelMA concentrations during 14 days of culturing. Scale bar: 200 µm. (**C**) Cell viability calculation at days 1, 7, and 14. (**D**) Cell diameter distributions in the liver lobule-like constructs with different GelMA concentrations after 14 days of cultivation.

**Figure 4 micromachines-14-00878-f004:**
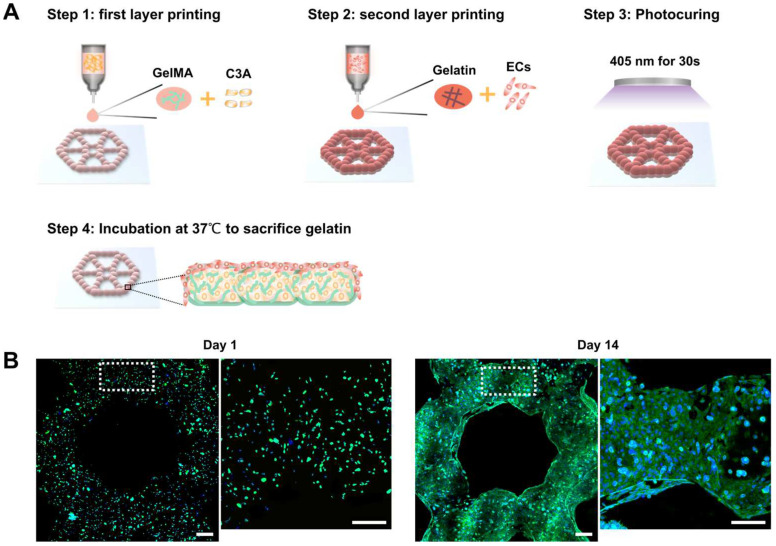
Endothelialized liver lobule-like constructs fabrication and culture. (**A**) Schematic illustrations of printing endothelialized liver lobule-like constructs in four steps. (**B**) F-actin staining of cells to show cell morphology at day 1 and day 14. Scale bar: 200 µm.

**Figure 5 micromachines-14-00878-f005:**
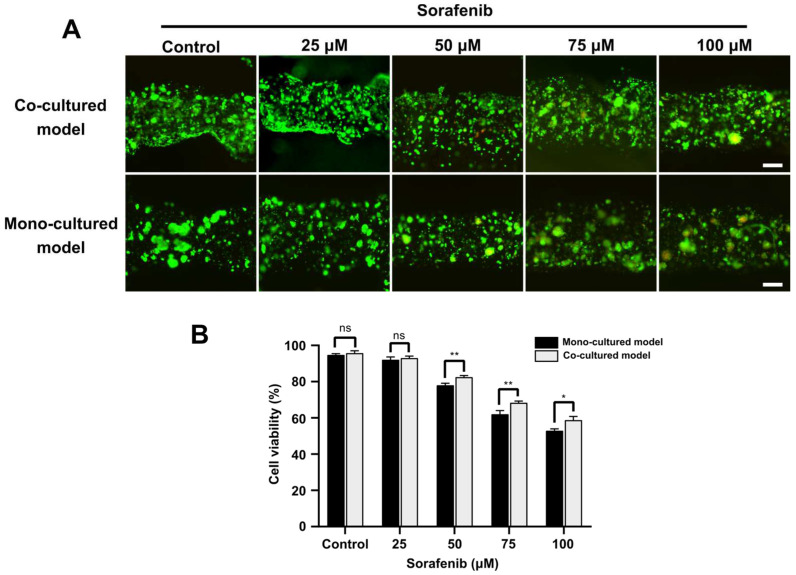
Drug evaluation on co-cultured liver cancer models and mono-cultured liver models. (**A**) Live/dead images showing different 3D liver cancer models after incubation with different concentrations of Sorafenib. Scale bar: 200 µm. (**B**) Statistical analysis of cell viabilities in both constructs at different drug concentrations. * *p* < 0.05 and ** *p* < 0.01.

## Data Availability

The data presented in this study are available on request from the corresponding author.

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
