# Peer review of "3D Bioprinting of an Endothelialized Liver Lobule-like Construct as a Tumor-Scale Drug Screening Platform"

_micromachines, 2023, doi:10.3390/mi14040878_

Round 1

Author Response

Summary: Here the authors have demonstrated the ability to use 3D bioprintng to construct functonal tssue assemblies of cancer organoids in GelMA hydrogel and endothelial cell layer. The in vitro tssue was then used to study the role of a drug, Sorafenib in cell viability. The authors show that cells clutered using multlayered 3D printng had increased viability to high concentratons of Sorafenib compared to that grown without an endothelium and when cultured as spheroids.

Comments: This is a very well writen manuscript that I enjoyed reading and I learned a few things about liver physiology as well. This method of culturing cells to develop high-throughput drug discovery pipeline seems promising and requires further optmizaton. The experiments were designed well and the conclusions are clear. However, I have a few misgivings regarding some of the experiments. I think a few more experiments need to be done before the paper can be considered for publicaton.

Response: Thank you very much for your valuable comments. According to your comments, we carefully revised the manuscript and marked all the changes using yellow background in the revised manuscript. Point-by-point responses to your comments are listed below here. We sincerely hope our revision could meet with your approval.

Major comments:

  1. F-actn staining is not enough to distinguish between the endothelial and cancer cells in the platorm. In Fig4., the authors should either use a fluorescently labelled HUVEC cell line or endothelial cell specifc antbody to isolate the role of the drug in cell viability and dynamics in HUVEC and liver cancer cells. This will help visualize the two cell layers and improve our understanding of why the cells are contributing more to the loss of vialibility.

Response: Thank you very much for your careful comment. In Fig.4, F-actin staining was conducted to characterize the morphology and spatial distribution of HUVECs on the surface of the C3A cells. So we did not further stain cells with specifc antbody. For drug screening assays, we have taken a common method to evaluate drug resistance by live/dead cell staining of all cells within the whole model. From the results, we found that the 3D endothelial network involved in the co-culture models collapsed as the concentration of the drug increased, and the obvious morphological differences between liver cancer cells (spheroids-like) and HUVECs (network-like) might help us for the observation, thanks.

  1. Quantfying the cell viability of huvec cells in isolaton will also help understand the barrier strength that the endothelial cells provide to improve the survivability of c3A cells.

Response: Thank you very much for this comment. In this work, cell viability was tested by live/dead kit. To be frank, it is hard to separately calculate the viability of HUVECs. We think the idea of measuring the viability of HUVECs in isolaton by using a fluorescently labelled HUVEC cell line is a new sight, which is valuable for our future work. We sincerely hope our revision could meet with your approval.

  1. It is not clear why the authors decided to conditon the cells for 48 hours with the drug. It would be interesting to see the temporal/longitudinal role of a partcular concentraton of Sorafenib on cell viability.

Response: Thank you very much for this comment. During drug screening assays, cell viability is usually evaluated within 24 to 72 h. Actually, we have measured cell viability at 24 h point, there was not a big difference between different 3D liver cancer models. We thus further treated models up to 48 h, and measured cell viabilities at a series of drug concentrations. Since we were mainly concerned the response of different 3D liver cancer models to different drug concentrations, we did not try longer treatment time. We feel so sorry that we did not have any longitudinal drug treatment results, and we will be happy to explore the longitudinal role of a partcular concentraton of Sorafenib on cell viability in our future work. Many thanks.

Minor comment:

In line 39, I think the authors mean species instead of ‘races’

Response: Thank you for your careful comment. We have replaced the term “races” by “species” in the revised manuscript. Many thanks.

Reviewer 2 Report

General:

The manuscript is generally difficult to read. It should definitely be revised again with regard to grammar. It contains many spelling mistakes, grammatically incorrect sentences and causal links that do not make sense in terms of content. It also contains abbreviations that are not explained at all or not at the first mention (e.g., C3A, PI, HepG2, FEP). Some sentences are unnecessarily long and should be split into several sentences (e.g., lines 188 - 192 or 333 - 337). With some phrases, it is not entirely clear to me what is meant, despite multiple readings. Some statements of the authors are not supported by the results shown.

Details:

There is a lack of information about the cells used; on the one hand, where they were obtained from, on the other hand, the information about what C3A cells are should also be in the paper.

The authors present porous structures in Figure 2 and discuss that these are pores in their hydrogels whose size depends on the gel concentration. They write: “Such structure can facilitate the penetration of nutrients and oxygen as well as providing a good local microenvironment for growing cells.” (line 237/238). These considerations are based on a fundamental misunderstanding. These Pores are generated during the freezing process by the generation and growth of water-ice crystals. These crystals displace and squeeze the hydrogel molecules during their growth. Each pore is generated by one crystal and the hydrogel molecules remain concentrated at the interfaces between the crystals. These pores do not exist in the hydrogel before freezing and, therefore, cannot affect “penetration of nutrients and oxygen”, “providing a good local microenvironment for growing cells”, or the formation of spheroids in the gel. The statement “The 6% GelMA showed the best performance to keep C3A cell spheroids grow, on the one hand possibly due to the large pores within the hydrogel inside favorable for cell proliferation.” (line 277-278), therefore, also makes no sense.

In Results and Discussion, the authors use evaluative terms such as "uniform" (line 183), "precise localization" (line 184), "effective" (line 184/185), "good shape fidelity" (line 192), which are neither defined nor substantiated. This can be accepted in the introduction, in results and discussion figures should be given. The "construct height" alone does not prove "good shape fidelity".

The authors write about "HUVECs differentiation" (line 75/76) and "the interaction between HUVECs and C3A cells" (line 314/315) without showing any studies on this. Can HUVECs be further differentiated?

The authors printed droplets (microbeads) with 700 µm diameter and "liver lobule constructs" with about 6-7 mm diameter. Natural liver lobules are usually around 1 up to 2 mm in diameter. The authors should point out this difference.

The authors claim "We successfully fabricated vascularized liver lobule constructs" (line 309) and "In this study, a 3D liver tumor model with vascularized liver lobule architecture was accurately constructed" (line 361/362). However, there are major differences between the structures of natural liver lobules and their printed structures not only in terms of size, but also in the spatial distribution of the different cells. Liver lobules have tubular vascular structures, the authors an endothelial coating on their structure. Thus, the printed constructs are not physiologically similar. The authors should discuss these differences.

The authors did use a cancer cell line. However, that alone does not make a "tumor model" (line 361). It is not clear what features of the printed structures the authors believe justify the terms "vascularized," "liver lobule construct," and "tumor model." The authors also use the term "tumor-scale" in the title and line 351; this is also not motivated.

In general, a scientific publication should also discuss the limitations of the validity of its own research results and the differences between its own models and the natural archetype.

Finally:

Why did the authors not actually try to print a liver lobule model with tubular vascularization? Actually, this should be possible with their technique.

Author Response

Response to Reviewer 2 Comments

General:

The manuscript is generally difficult to read. It should definitely be revised again with regard to grammar. It contains many spelling mistakes, grammatically incorrect sentences and causal links that do not make sense in terms of content. It also contains abbreviations that are not explained at all or not at the first mention (e.g., C3A, PI, HepG2, FEP). Some sentences are unnecessarily long and should be split into several sentences (e.g., lines 188 - 192 or 333 - 337). With some phrases, it is not entirely clear to me what is meant, despite multiple readings. Some statements of the authors are not supported by the results shown.

Response: Thank you very much for your valuable comments on our work. We have carefully revised the manuscript according to your suggestion. All the amendments have been marked using yellow background in the revised manuscript. We feel so sorry for the spelling or grammar mistakes, we have tried our best to refine the full text to avoid grammar or syntax errors, and a native English speaker was asked to polish our manuscript as well. Besides, abbreviations such as PI and FEP were explained at the first mention in the revised manuscript. As for C3A and HepG2, they are two kinds of human hepatocellular carcinoma cell lines, not abbreviations. Thank you again for your careful comments.

Details:

There is a lack of information about the cells used; on the one hand, where they were obtained from, on the other hand, the information about what C3A cells are should also be in the paper.

Response: Thanks for your careful comment. We have added more information about the origin of cells in the “2.5 Cell culturing” part. Many thanks.

The authors present porous structures in Figure 2 and discuss that these are pores in their hydrogels whose size depends on the gel concentration. They write: “Such structure can facilitate the penetration of nutrients and oxygen as well as providing a good local microenvironment for growing cells.” (line 237/238). These considerations are based on a fundamental misunderstanding. These Pores are generated during the freezing process by the generation and growth of water-ice crystals. These crystals displace and squeeze the hydrogel molecules during their growth. Each pore is generated by one crystal and the hydrogel molecules remain concentrated at the interfaces between the crystals. These pores do not exist in the hydrogel before freezing and, therefore, cannot affect “penetration of nutrients and oxygen”, “providing a good local microenvironment for growing cells”, or the formation of spheroids in the gel. The statement “The 6% GelMA showed the best performance to keep C3A cell spheroids grow, on the one hand possibly due to the large pores within the hydrogel inside favorable for cell proliferation.” (line 277-278), therefore, also makes no sense.

Response: Thank you very much for this comment. Actually, GelMA is a gelatin derivative with a majority of methacrylated groups. It is a commonly used functionalized biomaterial and bioink for 3D bioprinting because it is biocompatible, water-soluble and can be crosslinked with visiblelight to form a polymer network with tailorable physicochemical properties (Biomacromolecules 2000, 1, 31; Small 2021, 17, 2006050). Indeed, the freezing process during sample preparation has an effect on pore formation, while it is a common procedure for SEM characterization of hydrogel materials. Besides, the porous structures in GelMA hydrogel have been widely established and characterized through this manner, and results found that the pore size of the GelMA hydrogel is inversely related to the concentration. Many studies have demonstrated that a low concentration of GelMA is more appropriate for cellular functionalization and growth as it forms larger pores (Adv. Healthcare Mater. 2017, 6, 1601451; J Cell Mol Med 2021, 25, 880; Small 2019, 15, 1804216). For example, in the work of Xu et al. (J Cell Mol Med 2021, 25, 880), GelMA hydrogels with concentrations of 5%, 10%, and 15% showed different porous honeycomb structures, which can help to facilitate nutrient transport and provide space for proliferating and expanding encapsulated cells. We have supplemented the relevant references, thank you.

In Results and Discussion, the authors use evaluative terms such as "uniform" (line 183), "precise localization" (line 184), "effective" (line 184/185), "good shape fidelity" (line 192), which are neither defined nor substantiated. This can be accepted in the introduction, in results and discussion figures should be given. The "construct height" alone does not prove "good shape fidelity".

Response: Thank you very much for your valuable comment. Indeed, the dot extrusion printing (DEP) technology developed in our previous work (Mater Design 2022, 223, 111152) possesses the capacity of producing uniform hydrogel microbeads with precise localization. According to your comments, we have moved the corresponding content to the “Introduction” part in the revised manuscript. And, we have rephrased the "good shape fidelity", the corresponding sentence now reads as “indicating that the printed multi-layer structures could maintain good structure without microbeads collapse”. We sincerely hope our revision could meet with your approval.

The authors write about "HUVECs differentiation" (line 75/76) and "the interaction between HUVECs and C3A cells" (line 314/315) without showing any studies on this. Can HUVECs be further differentiated?

Response: Thank you very much for your insightful comment. We feel sorry for our less rigorous statement. Several previous studies have shown that when co-culturing HUVECs and C3A cells, cell-to-cell interactions were established through intercellular signaling molecules, such as cytokines and growth factors. In our work, we mainly focused on the construction of the endothelial network, truly have no results on cell interactions. We thus have revised the corresponding sentence as follows ”adjacent HUVECs were interconnected, ultimately elongating and branching to form a layer of vascular network structure on the surface of the construct”. Thank you again.

The authors printed droplets (microbeads) with 700 µm diameter and "liver lobule constructs" with about 6-7 mm diameter. Natural liver lobules are usually around 1 up to 2 mm in diameter. The authors should point out this difference.

Response: Thank you very much for this comment. According to your comments, we have added content to point out the difference in “Conclusion” part. Many thanks.

The authors claim "We successfully fabricated vascularized liver lobule constructs" (line 309) and "In this study, a 3D liver tumor model with vascularized liver lobule architecture was accurately constructed" (line 361/362). However, there are major differences between the structures of natural liver lobules and their printed structures not only in terms of size, but also in the spatial distribution of the different cells. Liver lobules have tubular vascular structures, the authors an endothelial coating on their structure. Thus, the printed constructs are not physiologically similar. The authors should discuss these differences.

Response: Thank you very much for this insightful comment. In this work we utilized DEP technology to fabricate 3D liver cancer models for anti-tumor drug screening. Indeed, the bioprinted 3D liver cancer models with lobule-like morphology could not well replicate the structures of natural liver lobules. Thus, we have rewrote the “liver lobule” as “liver lobule-like” as other studies (Small 2020, 1905505; Molecules 2019, 24, 1762; Cells 2021, 10, 1268) commonly described, and the relevant textual description was modified in the revised manuscript. Aslo, we have added some discussion about the difference, thank you again.

The authors did use a cancer cell line. However, that alone does not make a "tumor model" (line 361). It is not clear what features of the printed structures the authors believe justify the terms "vascularized," "liver lobule construct," and "tumor model." The authors also use the term "tumor-scale" in the title and line 351; this is also not motivated.

Response: Thank you very much for this comment. Actually, a cancer cell line is indeed a simple tumor model. Beyond 2D-culture models and animal models, 3D in vitro tumor models such as multicellular spheroids, cell-seeded scaffolds, and cell-laden hydrogel constructs have been widely studied. In this work, we bioprinted a liver lobule-like construct for co-culturing C3A cells and HUVECs, providing one kind of 3D liver cancer models for drug screening. As for the term "vascularized", we feel sorry for our less rigorous description. We thus replaced it with “endothelialized”. Finally, here, compared to multicellular spheroids at small scale, the proposed 3D liver cancer model with a millimeter-sized structure provided biologically-relevant size, which might advance preclinical drug screening capacities.

In general, a scientific publication should also discuss the limitations of the validity of its own research results and the differences between its own models and the natural archetype.

Response: Thank you very much for this comment. According to your comments, we have added more discussion about the results in the “Conclusion” part. We sincerely hope our revision could meet with your approval.

Finally:

Why did the authors not actually try to print a liver lobule model with tubular vascularization? Actually, this should be possible with their technique.

Response: Thank you very much for your valuable comment. Indeed, by utilizing the extrusion printing, micro-channel structures can be fabricated by inclusion of sacrificial materials. In particular, the tubular vascular structures can also be achieved by simultaneous extrusion of two different bioinks through coaxial nozzles, forming strands with a core and a shell compartment. In our future work, we will be happy to fabricate liver lobule models with tubular vascularization for better replicating the structural features of liver tissues.

Round 2

Reviewer 1 Report

Basically there were no new experiments done based on my suggestions. If new experiments don't fit the scope of the journal, the authors should at least do the following:

1. Use confocal microscopy to take the images of the printed constructs and delineate the boundary between huvec cells  and C3A cells. The images right now as they are don't clearly indicate if its endothelial cells on top of the liver lobule or side by side. 

2. The methods section should clearly mention what kind of fluorescent microscopy was conducted to get the images. 

Author Response

Basically there were no new experiments done based on my suggestions. If new experiments don't fit the scope of the journal, the authors should at least do the following:

Response: Thank you very much for your careful comments. We are sorry that we did not supplement the experiment last time due to lack of time. Here, we provided confocal photographs of the endothelialized liver lobule-like construct to make the structure more clearly. Sincerely hope our revision could meet with your approval.

  1. Use confocal microscopy to take the images of the printed constructs and delineate the boundary between huvec cells and C3A cells. The images right now as they are don't clearly indicate if its endothelial cells on top of the liver lobule or side by side.

Response 1: Thank you very much for this comment. We have scanned the endothelialized liver lobule-like construct from top to bottom by using a confocal microscope, and the result showed that endothelial cells were on top of the liver lobule. We have included the result as Video S1. Thanks.

  1. The methods section should clearly mention what kind of fluorescent microscopy was conducted to get the images.

Response 2: Thank you very much for your valuable comment. We have added information about the fluorescent microscopy in the “2.9 Cell morphology analysis” part in the revised manuscript. Many thanks.

Reviewer 2 Report

Now, the manuscript is much better, in terms of grammar and readability, but also in terms of content. However, I still have to repeat two criticisms:

1) The authors still claim in their manuscript that the pores shown in Figure 2 would exist like this in their bulk GelMA hydrogel. However, this is wrong. These pores are only formed during freeze-drying. That this is the case has been shown often enough in the scientific literature: This effect is exploited by many researchers to produce scaffolds. In this context, the pore size can be determined, in addition to the concentration of the starting gel, by parameters of the freeze-drying process, for example, temperature (e.g.: Grenier et al, Mechanisms of pore formation in hydrogel scaffolds textured by freeze-drying, Acta Biomaterialia 94, 195-203 (2019) doi: 10.1016/j.actbio.2019.05.070). The size of these pores corresponds to the size of the ice crystals formed during freezing. The size of the cross-linked networks in the bulk hydrogel is much smaller, about 100 nanometers or smaller. In contrast, the pores shown by the authors are about 100 microns in size, 1000 times larger.

The microbeads the authors printed are 700 microns in diameter! Do the authors think such a microbead consists of 100-µm pores? So, 7 pores in a row over the diameter? Then their microbeads should look more like foam or a piece of sponge instead of having a smooth surface. Moreover, it should be possible to visualize these pores without freeze-drying, at least indirectly, for example by confocal microscopy with non-clustering cells in individual microbeads.

No, these pores do not exist in the hydrogel before freeze-drying. The authors' argument that freeze-drying is a common method for studying hydrogels is not contradictory here. One can infer properties of the hydrogel from properties of the freeze-dried gel, but one must not simply equate them.

In the manuscript, the authors present it as if there are these pores in the hydrogel, and try to explain observations with these pores. But this is wrong. That this error can also be found in other publications is sad, but does not change the facts.

2) The authors use the term "tumor-scale" several times, including in the title. However, it is probably unclear to a broader readership (like that of Micromachines) not specialized in cancer research what is meant by this. There is no statement, whether liver tumors are usually as large as the printed constructs, i.e., 6 - 7 mm in diameter; or is there some other reason to speak of "tumor-scale"? The use of this term should be briefly explained and motivated.

Minor comment:

It would be nice if the authors could put the word "hepatocytes" or something similar in the abstract instead of just writing "C3A".

Author Response

Now, the manuscript is much better, in terms of grammar and readability, but also in terms of content. However, I still have to repeat two criticisms:

1) The authors still claim in their manuscript that the pores shown in Figure 2 would exist like this in their bulk GelMA hydrogel. However, this is wrong. These pores are only formed during freeze-drying. That this is the case has been shown often enough in the scientific literature: This effect is exploited by many researchers to produce scaffolds. In this context, the pore size can be determined, in addition to the concentration of the starting gel, by parameters of the freeze-drying process, for example, temperature (e.g.: Grenier et al, Mechanisms of pore formation in hydrogel scaffolds textured by freeze-drying, Acta Biomaterialia 94, 195-203 (2019) doi: 10.1016/j.actbio.2019.05.070). The size of these pores corresponds to the size of the ice crystals formed during freezing. The size of the cross-linked networks in the bulk hydrogel is much smaller, about 100 nanometers or smaller. In contrast, the pores shown by the authors are about 100 microns in size, 1000 times larger.

The microbeads the authors printed are 700 microns in diameter! Do the authors think such a microbead consists of 100-µm pores? So, 7 pores in a row over the diameter? Then their microbeads should look more like foam or a piece of sponge instead of having a smooth surface. Moreover, it should be possible to visualize these pores without freeze-drying, at least indirectly, for example by confocal microscopy with non-clustering cells in individual microbeads.

No, these pores do not exist in the hydrogel before freeze-drying. The authors' argument that freeze-drying is a common method for studying hydrogels is not contradictory here. One can infer properties of the hydrogel from properties of the freeze-dried gel, but one must not simply equate them.

In the manuscript, the authors present it as if there are these pores in the hydrogel, and try to explain observations with these pores. But this is wrong. That this error can also be found in other publications is sad, but does not change the facts.

Response: Thank you very much for this valuable comment. We have understood what the reviewer meant. Indeed, the freeze-drying process generated porous structure in the GelMA hydrogel. We feel sorry for our less rigorous statement. Thus, we modified the corresponding part, and marked the content in yellow background. Also, some discussion about the pores have been removed, thank you again.

2) The authors use the term "tumor-scale" several times, including in the title. However, it is probably unclear to a broader readership (like that of Micromachines) not specialized in cancer research what is meant by this. There is no statement, whether liver tumors are usually as large as the printed constructs, i.e., 6 - 7 mm in diameter; or is there some other reason to speak of "tumor-scale"? The use of this term should be briefly explained and motivated.

Response: Thank you very much for this comment. In general, 3D cultured hepatic cancer spheroids have been widely used in the screening of anti-tumor drugs in recent years. While these spheroids are usually on the scale of several hundred micrometers, which is unrealistic compared to the size of native tumors. Thus, in this work, we tried to develop a 3D liver cancer model with a millimeter-sized structure to provide biologically-relevant size, which might advance preclinical drug screening capacities. The relevant content has been described in “Introduction” part, thanks.

Minor comment:

It would be nice if the authors could put the word "hepatocytes" or something similar in the abstract instead of just writing "C3A".

Response: Thank you very much for this comment. We have replaced "C3A" by "hepatocytes" in the abstract. Many thanks.

Round 3

Reviewer 2 Report

Comments on Response to Reviewer 2 Comments

[…]

Authors' Response: Thank you very much for this valuable comment. We have understood what the reviewer meant. Indeed, the freeze-drying process generated porous structure in the GelMA hydrogel. We feel sorry for our less rigorous statement. Thus, we modified the corresponding part, and marked the content in yellow background. Also, some discussion about the pores have been removed, thank you again.

Reviewer's Comment: The authors would still have to adjust lines 278-280 here:
“The 6% GelMA showed the best performance to keep C3A cell spheroids grow, on the one hand possibly due to the large pores within the hydrogel inside favorable for cell proliferation [41,43].”

2) The authors use the term "tumor-scale" several times, including in the title. However, it is probably unclear to a broader readership (like that of Micromachines) not specialized in cancer research what is meant by this. There is no statement, whether liver tumors are usually as large as the printed constructs, i.e., 6 - 7 mm in diameter; or is there some other reason to speak of "tumor-scale"? The use of this term should be briefly explained and motivated.

Authors' Response: Thank you very much for this comment. In general, 3D cultured hepatic cancer spheroids have been widely used in the screening of anti-tumor drugs in recent years. While these spheroids are usually on the scale of several hundred micrometers, which is unrealistic compared to the size of native tumors. Thus, in this work, we tried to develop a 3D liver cancer model with a millimeter-sized structure to provide biologically-relevant size, which might advance preclinical drug screening capacities. The relevant content has been described in “Introduction” part, thanks.

Reviewer's Comment: The content I found in the introduction, line 48-51, is:

[…] spheroids are limited to small-scale models […]. By combining cells and hydrogel through 3D bioprinting technology, 3D liver cancer models with more complex structures and tumor-scale microenvironments can be constructed [18-20].

There is no information on how large liver tumors usually are, at what size one can speak of "tumor-scale" and what advantages such a tumor-scale model actually has. Since tumors usually start as single cells and can grow to a size of more than ten centimeters, they are at some time as large as "spheroids". Unfortunately, the term "tumor-scale" is neither defined nor motivated in the manuscript.

Here, in their "Response to Reviewer 2 Comments", the authors at least give a size specification for "spheroids". Here they should also give a size for "size of native tumors" for liver tumors and write 1-2 sentences why this difference in size is relevant for "preclinical drug screening" - in the manuscript and not in Response to Reviewer. Thus, it could be justified to use this term in the title and to emphasize it so strongly in the introduction.

Author Response

  1. Authors' Response: Thank you very much for your comment. We have modified “The 6% GelMA showed the best performance to keep C3A cell spheroids grow, on the one hand possibly due to the large pores within the hydrogel inside favorable for cell proliferation [41,43].” in the revised manuscript.
  2. Authors' Response: Thank you very much for your comment. We have included the content in the “Introduction” part, thanks.